# Experimental Study on Effects of Adjustable Vaned Diffusers on Impeller Backside Cavity of Centrifugal Compressor in CAES

Zhihua Lin [1,2], Zhitao Zuo [1,2,3], Wenbin Guo [1,2], Jianting Sun [1], Qi Liang [1] and Haisheng Chen [1,2,3,4,*]

1    Institute of Engineering Thermophysics, Chinese Academy of Sciences, Beijing 100190, China; linzhihua@iet.cn (Z.L.); zuozhitao@iet.cn (Z.Z.); guowenbin@iet.cn (W.G.); sunjianting@iet.cn (J.S.); liangqi@iet.cn (Q.L.)

2    University of Chinese Academy of Sciences, Beijing 100049, China

3    National Energy Large Scale Physical Energy Storage Technologies R&D Center of Bijie High-Tech Industrial Development Zone, Bijie 551712, China

4    Nanjing Institute of Future Energy System, Institute of Engineering Thermophysics, Chinese Academy of Sciences, Nanjing 211135, China

*    Correspondence: chen_hs@iet.cn; Tel.: +86-010-82543148

**Abstract:** The impeller backside cavity (IBC) is a unique structure of centrifugal compressor in compressed air energy storage (CAES) systems, which affects the aerodynamic performance of centrifugal compressor, and the angle change of the downstream coupled adjustable vaned diffusers (AVDs) will affect the flow field inside the cavity and compressor performance. This paper relies on the closed test facility of the high-power intercooling compressor to measure static pressure and static temperature at different radii on the static wall of the IBC. The coupling relationship between the IBC and compressor under variable operating conditions is analyzed, and the influence of AVDs on the internal flow in IBC is studied. The results show that static pressure and static temperature rise along the direction of increasing radius, but static temperature drops near the coupling between the impeller outlet and the cavity inlet. Under AVDs' design angle, static pressure and static temperature at each point, static pressure loss and static temperature loss in the direction of decreasing radius all increase as the flow decreases. Under variable AVDs' angles, static pressure and static temperature will change differently, and respective loss will also be different.

**Keywords:** centrifugal compressor; impeller backside cavity; adjustable vaned diffusers; variable operating conditions

## 1. Introduction

    In the field of distributed energy applications with fluctuant and intermittent renewable energy (photovoltaics [1,2], wind power [3,4]) and the construction of smart grids and microgrids, compressed air energy storage (CAES) system is considered a key technology to improve penetration of renewable energy and provide important support for peak power regulation [5,6]. The centrifugal compressor is the core component of the system and its energy conversion efficiency will directly affect the overall efficiency and economy [7]. There is an impeller backside cavity (IBC) structure in the centrifugal compressor, which has an important influence on the flow field details, pressure ratio, efficiency, torque, shaft power, and axial thrust of the centrifugal compressor [8–11].

    In essence, the IBC is a limited annular space between rotating disk and static wall, which has a unique flow field structure and flow characteristics that are different from the mainstream, resulting in friction loss and leakage flow loss. Its outer edge junction surface is connected with the mainstream of compressor, and the inner edge junction surface is connected with an air seal. It is mainly affected by the movement of the impeller, the heat transfer efficiency of both side walls, the different types of boundary layers on both side walls, the existence and type of the rotating core between the boundary

layers on both sides, the mainstream pressure and flow coefficient for gas exchange at the outer edge interface. Simultaneously, as a variable-condition adjustment technology and the downstream coupled components of the IBC, adjustable vaned diffusers (AVDs) can expand the working flow range of the centrifugal compressor [12] and affect the flow field inside the cavity [13,14], Cravero [15,16] also provided a particular flow mechanism at the diffuser inlet (near the back cavity inlet). It's significant to investigate the aerodynamic parameter distribution in the IBC and the coupled interaction between the IBC and AVDs, which contributes to analyzing the internal flow field structure and loss distribution of the centrifugal compressor in CAES.

Depending on the working conditions and the type of air seal, there may or may not be fluid flow at the inner edge interface of IBC. In the study of the simplified cavity flow between a finite rotating and stationary disk, most of them are based on incompressible viscous working media between parallel walls. In the cavity flow field without radial flow, Daily [17] and Nece [18] pointed out that there are four different flow modes in the cavity and studied the roughness effects on frictional resistance. Circular and spiral waves were observed inside the cavity by Schouveiler [19]. Serre [20] and Séverac [21] researched the flow transition and turbulence between a rotating and stationary disk. Farthing [22,23] and Bohn [24,25] explained many flow and heat transfer phenomena inside the rotating cavity with an axial throughflow of cooling air in experiments. Alexiou [26], Owen [27,28] and Pitz [29] analyzed buoyancy-induced flow in a rotating cavity. In the cavity flow field with radial flow, Poncet [30,31], Rémy [32,33] and Debuchy [34,35] investigated the cavity flows with superimposed flows. Coren [36] extended the range of data available for windage in rotor-stator systems. Luo [37] and Tao [38] experimentally summarized the empirical relationships of the dimensionless wind resistance temperature rise parameter and the dimensionless wind resistance torque coefficient in the cavity.

In the research of the actual cavity of the impeller disk, there are few experimental results, especially in the centrifugal compressor. Gantar [39] measured the pressure and velocity distributions inside the cavity of a centrifugal pump, and studied the effect of radial flow on the axial thrust of the impeller. Gulich [40] proposed a general method to predict the disk friction loss of closed turbomachine impellers. Sun [8] and Li [41] numerically studied the flow field structures of IBC and its influences on the centrifugal compressor. The cavity sealing flow in deeply scalloped radial turbines was numerically investigated by He [42]. Zeng [43] simulated three-dimensional coupled flow of secondary air system and main flow passages in a micro gas turbine. Shahin [44] investigated the relative effect of cavity on flow characteristics and sound levels through the use of Large Eddy Simulation and Ffowcs Williams–Hawkings model. Marechale [45], Qiao [46] and Hazby [47] evaluated the effect of labyrinth seal clearances on the compressor impeller performance. Dong [48,49] studied the radial distributions of the dimensionless tangential and radial velocities in the back shroud and hub cavities of a centrifugal pump and the influences of the diameter of the balance hole. Liu [50] numerically proposed two methods of baffle and swirl-controlled orifice to regulate the pressure loss and distribution in the cavity with various radial inflow (including $Re_\omega$, $C_w$, $G$, $\beta_0$, $\lambda_T$, etc.).

However, according to the above published documents, the simplified cavity model cannot truly reflect the actual flow field structure in the IBC, most of the numerical results in impeller disk are rarely verified by experiments and the universality of measurement results in pump is limited for compressor. Particularly, there is a lack of literature on the IBC in the centrifugal compressor for CAES, in which experimental analysis is even more urgent. In this paper, relying on the large-scale CAES centrifugal compressor closed test rig, the static pressure and static temperature at different radii on the static wall of IBC are measured in detail. The changes in the distributions of aerodynamic parameters in the cavity under different mainstream flow and the influence of the cavity on the compressor performance are analyzed. Considering the influence of the downstream component of AVDs, the coupling characteristics of the whole machine and the change law of the flow field in the cavity under different angles of AVDs are studied.

## 2. Research Object

As shown in Figure 1, The research object is the low-speed and low-pressure centrifugal compressor in the multi-stage intercooling closed test facility. The main flow components of the centrifugal compressor include 12 adjustable inlet guide vanes (AIGVs), 13 semi-open impeller blades, 11 low-solidity adjustable vaned diffusers (AVDs) and an asymmetric circular volute. IBC is composed of an impeller wheel disk, a casing, a rotating shaft, and a sealing structure, which has been reduced in weight from the perspective of rotor dynamics during the casting process. The interface of the outer edge is connected with the mainstream of the impeller outlet, and the interface of the inner edge is connected with the carbon ring seal and the oil seal. The sealing effect is good, and the relative leakage is very small. The design parameters of the compressor are as follows: total inlet pressure 97,000 Pa, total inlet temperature 303.15 K, design mass flow 34 kg/s, design rotating speed 8658.7 r/min, total pressure ratio 2.29 with consideration of volute. The gap ratio of the simplified dynamic and static disk model corresponding to the measuring point at the maximum radius of the cavity is 0.036, the main structural parameters of the impeller and AVDs are shown in Table 1.

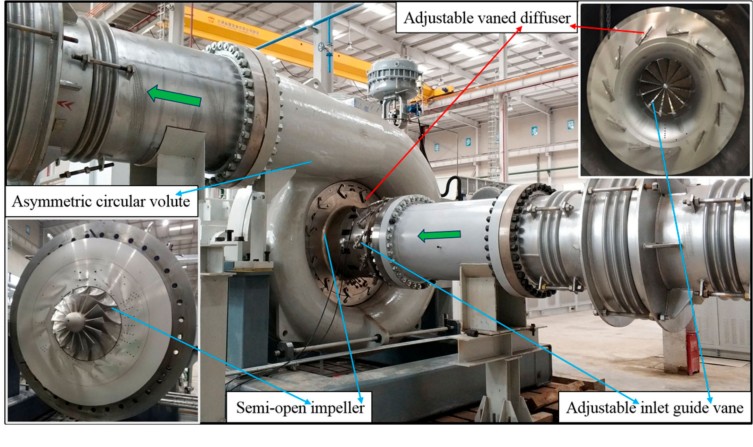

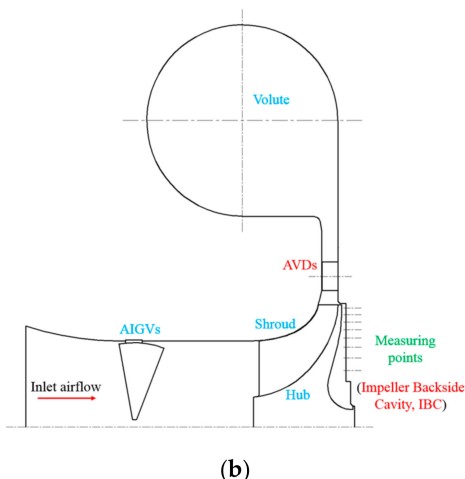

(**a**)                                                                 (**b**)

**Figure 1.** Site map and geometric structure of the research object: (**a**) the experimental compressor and main components; (**b**) the geometry of stage and IBC.

**Table 1.** Main parameters of impeller and AVDs.

| Design Parameters | Impeller | AVDs |
|---|---|---|
| Tip clearance | 1.17 mm | - |
| Blade inlet radius | 294.4 mm | 466.2 mm |
| Inlet installation angle | 30° | 27° |
| Inlet vane span | 190.0 mm | 55.4 mm |
| Blade outlet radius | 417.5 mm | 564.3 mm |
| Outlet installation angle | 60° | 31° |
| Outlet vane span | 65.9 mm | 55.4 mm |

## 3. Test System and Measuring Equipment

### 3.1. Test Facility and Measuring Points

This test work was completed on the closed test rig of the high-power multi-stage intercooling centrifugal compressor of the National Energy Large-scale Physical Energy Storage Technologies R&D Center (Institute of Engineering Thermophysics, Chinese Academy of Sciences). The facility has strong capabilities with maximum 16,000 r/min output speed and maximum 5200 kW transmission power, the mechanical transmission efficiency is above 95.0%. Its driveline consists of a GH180 Medium-Voltage inverter, a 5.2 MW AC inverter motor, and a 16,000/2200 revolutions per minute (RPM) increasing gearbox to

drive the impeller. Furthermore, the speed of Variable-frequency Drive (VFD) adopts closed-loop control, the control accuracy is better than 0.05%, the AC inverter motor speed adjustment range is 150–2200 r/min. In addition, the low-speed ratio gearbox adopts double-split type with maximum 4700 N·m output torque, the accuracy of the low-speed and high-speed non-contact torque measuring devices on both sides of the gearbox are both 0.3%.

According to the equal area method, eight static pressure measuring points and eight static temperature measuring points are densely arranged on the static wall surface of IBC, and the radii of measuring points are determined to be 193.5, 236.2, 272.1, 303.8, 332.4, 358.9, 383.3 and 406.3 mm. Each radius has a pressure measuring hole and a temperature measuring hole, the specific measuring point positions are shown in Figure 2a. The 16 measuring points are respectively led out to the circumference of the impeller backside with a radius of 535 mm through the corresponding internal channels. The eight marked points A, B, C, D, E, F, G and H on the right are the lead-out positions of the eight pressure measuring holes. The eight marked points I, J, K, L, M, N, P and Q on the left are the lead-out positions of the 8 temperature measuring holes. The specific correspondence is shown in Table 2 and Figure 3, which also includes the gap ratio at each measuring point in the corresponding simplified dynamic and static disk model. The pressure tubes are connected to the remote PSI pressure scanning system through the eight marked points on the right, and the thermal resistance lead wires are connected to the remote temperature transmitter through the eight marked points on the left, which realize the remote transmission of pneumatic parameters in the measurement site of IBC as Figure 2b showed.

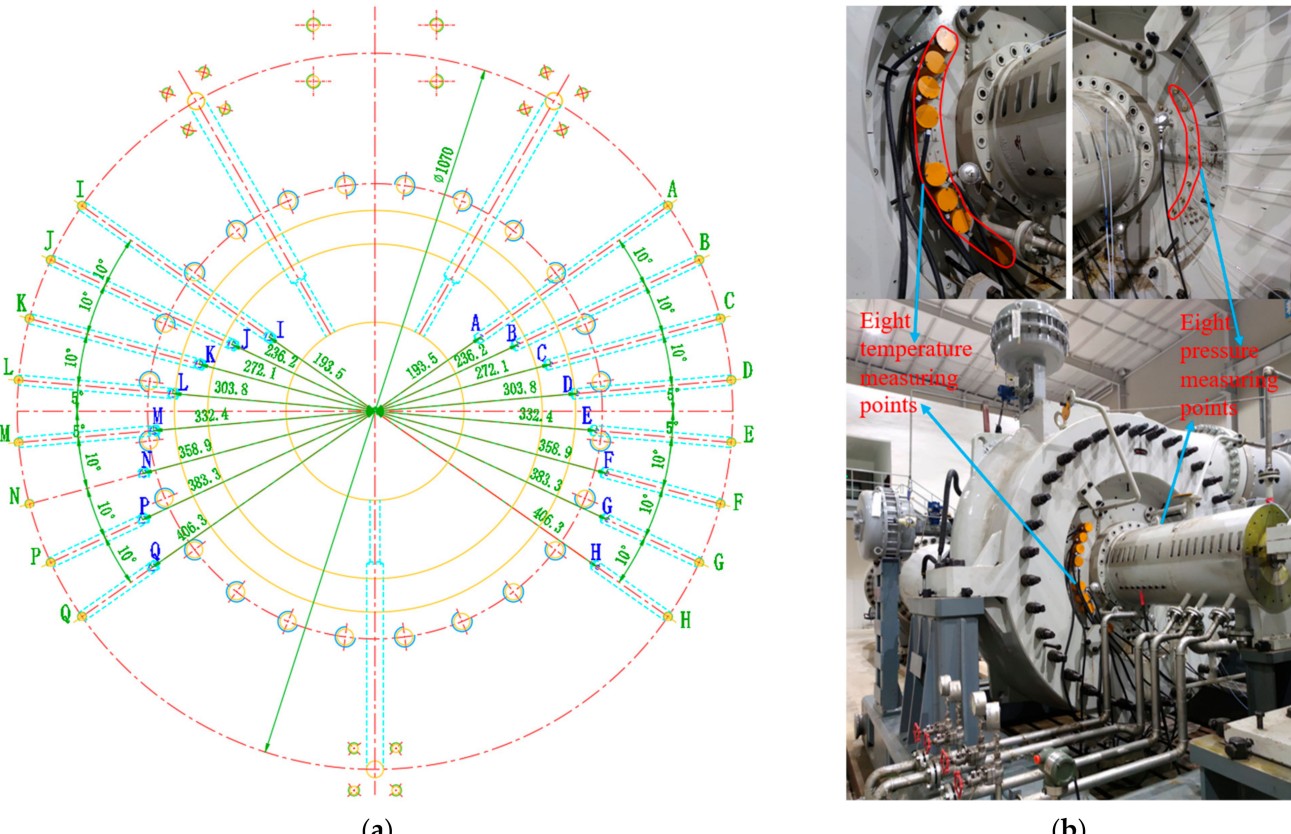

(a)  (b)

**Figure 2.** Specific location of measuring points and measurement site: (**a**) layout of the measuring points of IBC; (**b**) measurement site of IBC.

**Table 2.** Specific correspondence of measuring points and marked points.

| Radius (mm) | 193.5 | 236.2 | 272.1 | 303.8 | 332.4 | 358.9 | 383.3 | 406.3 |
|---|---|---|---|---|---|---|---|---|
| Static pressure measuring points | A | B | C | D | E | F | G | H |
| Static temperature measuring points | I | J | K | L | M | N | P | Q |
| Axial distance of cavity (mm) | 15.00 | 15.00 | 15.35 | 17.65 | 22.59 | 30.69 | 40.30 | 51.77 |
| Gap ratio | 0.036 | 0.036 | 0.037 | 0.042 | 0.054 | 0.074 | 0.097 | 0.124 |

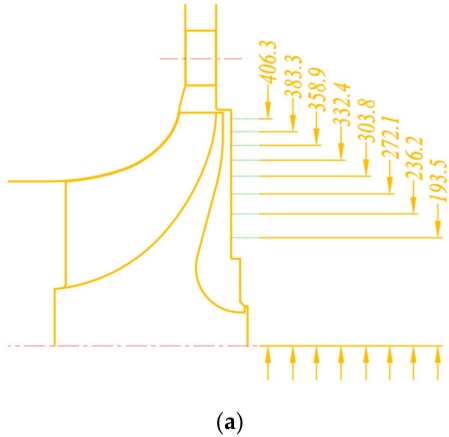

(**a**)

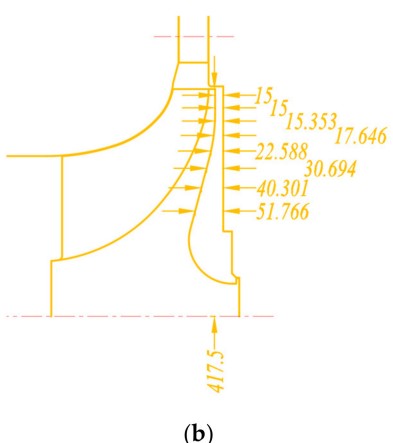

(**b**)

**Figure 3.** Zoom figure of the measuring points in IBC: (**a**) measuring point radius; (**b**) axial clearance of measuring point.

### 3.2. Coupling Relationship between AVDs and IBC

The diffuser is a downstream component of the outer edge interface of IBC. The high-entropy fluid in the cavity is mixed with the mainstream, which directly increases the total pressure loss in the diffuser, especially in the leading edge and backside of the blade [13]. Simultaneously, a local acceleration zone appears at the inlet of the diffuser, which will run through the entire diffuser passage, resulting in a decrease in compressor performance. The diffuser also affects the upstream cavity flow field structure, and different inlet installation angles of AVDs' blade will have different effects on the internal flow of the cavity.

In this research, five diffuser angles ($-8°$, $-4°$, $0°$, $+4°$ and $+8°$) were selected through AVDs to study the coupling characteristics with IBC under variable diffuser angle. The diffuser angle refers to the rotation angle of AVDs around their own rotation axis, which is achieved through a complex mechanical transmission mechanism (including pneumatic diaphragm actuators, electric control valves, pull rods, connecting rods, rotating discs, pull plates, transmission shaft, etc.). The zero angle corresponds to AVDs' design inlet installation angle of $27°$. The positive angle corresponds to the increase of the inlet installation angle, the distance between the leading edge of the blade and the outer edge of the cavity is shortened, and the coupling relationship is closer. The situation of the negative angle is opposite to the above and the specific angle correspondence is shown in Table 3.

**Table 3.** Corresponding relationship between the AVDs' angle and inlet installation angle of blade.

| AVDs' angle | $-8°$ | $-4°$ | $0°$ | $+4°$ | $+8°$ |
|---|---|---|---|---|---|
| Inlet installation angle | $19°$ | $23°$ | $27°$ | $31°$ | $35°$ |

### 3.3. Measurement Method and Error Analysis

The overall structure of the entire test system can be roughly divided into three layers. The first layer is the field data collection and the feedback of the control valve signal, including total pressure probe combs and WRN thermocouple total temperature probe combs for inlet and outlet pipes, UNIK5000 pressure sensors and WZPK thermal resistances PT100 between the flow parts, annubar flowmeter, vibration sensor, non-contact torque meter and valve opening. The second layer is the data processing and control system, which uses the PLC data acquisition management system to process, analyze, alarm and

complete related adjustment and control of the sensor feedback signal. The third layer is remote monitoring, which uses transmission media such as optical fiber and twisted pair to transmit the processed signal to the host computer. For the measurement of static pressure and static temperature measurement points at different radii on the static wall of IBC, Model 9216 Pneumatic Intelligent Pressure Scanners are selected to directly transmit the static pressure signal to PSI9000 pressure acquisition system through the pressure pipe. Three-wire embedded thermal resistance, thermal resistance temperature transmitter, and PLC data acquisition module are selected to connect the static temperature signal to the host computer. The relevant parts of the measurement equipment are shown in Figure 4.

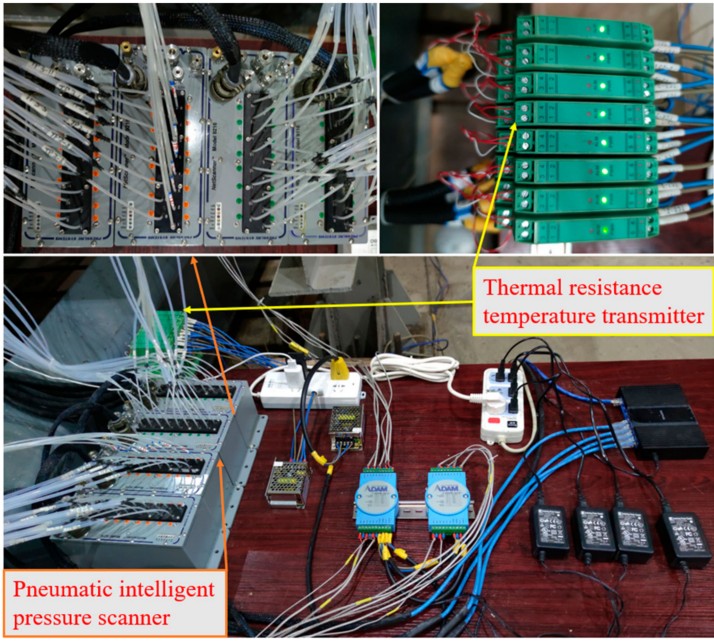

**Figure 4.** Measuring equipment of IBC.

During the test, the inlet pressure and temperature can be controlled and held constant independently of ambient conditions for ensuring accuracy and repeatability. The back pressure was established by adjusting the exhaust throttle valve group, so as to perform the performance curve test. However, due to factors such as shaft seal air leakage, the back pressure adjustment process often requires opening the supplemental valve or the bleed valve for coordination, resulting in fluctuations in the pneumatic parameters. The fluctuation deviation conforms to the allowable range of ASME PTC-10 standard [51], and the specific values are shown in Table 4.

**Table 4.** Permissible fluctuations of experimental data.

| Parameter | Inlet Temperature | Inlet Pressure | Rotating Speed | Shaft Power | Mass Flow |
| --- | --- | --- | --- | --- | --- |
| Deviation | 0.5% | 2.0% | 0.5% | 1.0% | 2.0% |

The recorded physical quantities mainly include $p_t$, $T_t$, $m$, $n$, $T$, $p_s$, $T_s$ and other physical quantities can be obtained through data processing and thermodynamic calculations, such as $PR_t$, $\eta_{is}$, $P_s$.

$$PR_t = \frac{p_{t,out}}{p_{t,in}} \tag{1}$$

$$\eta_{is} = \frac{T_{t,in}\left[(p_{t,out}/p_{t,in})^{(\gamma-1)/\gamma} - 1\right]}{T_{t,out} - T_{t,in}} \tag{2}$$

$$P_s = \frac{2\pi n T}{60 \times 10^3} \tag{3}$$

When accidental errors and gross errors are not considered, the relative errors of experimental measurement parameters are caused by the accuracy of the instruments and sensors, the specific values are shown in Table 5.

**Table 5.** The accuracy of sensors.

| Sensor Name | Accuracy |
|---|---|
| WRN thermocouple total temperature probe comb | $\pm(0.15 + 0.004\,|\,\mathrm{t}\,|)$ |
| WZPK thermal resistance PT100 | $\pm(0.15 + 0.002\,|\,\mathrm{t}\,|)$ |
| Total pressure probe comb | $\pm 0.05\%$ FS |
| UNIK5000 pressure sensor | $\pm 0.04\%$ FS BSL |
| Manner speed sensor | $\pm 0.05\%$ FS $\pm 1$ r/min |
| Manner non-contact torque sensor | $\pm 0.3\%$ |
| Annubar flow Sensors | $\pm 1.5\%$ |
| Static temperature sensor of IBC | $\pm 0.1\%$ FS |
| Static pressure sensor of IBC | $\pm 0.05\%$ FS |

In addition, the relative errors of $PR_t$, $\eta_{is}$, $P_s$ are obtained from the error of the direct measurement value through the error transfer formulas as followed.

$$\frac{d\overline{y}}{y} = \sqrt{\left(\frac{x_1}{y}\frac{\partial y}{\partial x_1}\frac{dx_1}{x_1}\right)^2 + \left(\frac{x_2}{y}\frac{\partial y}{\partial x_2}\frac{dx_2}{x_2}\right)^2 + \cdots + \left(\frac{x_n}{y}\frac{\partial y}{\partial x_n}\frac{dx_n}{x_n}\right)^2} \tag{4}$$

Derived from Equation (4), the maximum error equations of $PR_t$, $\eta_{is}$, $P_s$ are as follows:

$$\frac{d\overline{PR_t}}{PR_t} = \sqrt{\left(\frac{dp_{t,in}}{p_{t,in}}\right)^2 + \left(\frac{dp_{t,out}}{p_{t,out}}\right)^2} \tag{5}$$

$$\frac{d\overline{\eta_{is}}}{\eta_{is}} = \sqrt{\left(\frac{dT_{t,in}}{T_{t,in}}\right)^2 + \left(\frac{dDT_t}{DT_t}\right)^2 + A^2\left(\frac{dp_{t,in}}{p_{t,in}}\right)^2 + A^2\left(\frac{dp_{t,out}}{p_{t,out}}\right)^2}$$
$$A = \left(\frac{\gamma-1}{\gamma}\right)\bigg/\left[1 - (p_{t,out}/p_{t,in})^{(1-\gamma)/\gamma}\right] \tag{6}$$
$$d\Delta T_t = dT_{t,in} + dT_{t,out}$$

$$\frac{d\overline{P_s}}{P_s} = \sqrt{\left(\frac{dn}{n}\right)^2 + \left(\frac{dT}{T}\right)^2} \tag{7}$$

Calculated by Equations (5)–(7), the error ranges of $PR_t$, $\eta_{is}$, $P_s$ are $\pm 0.07\%$, $\pm 0.75\%$, and $\pm 0.31\%$ respectively, which is acceptable in experimental analysis.

## 4. Results and Discussion

### 4.1. Aerodynamic Parameters Distributions in IBC

Because the cavity has a special geometric structure such as in Figure 1b, the gap ratio at each measuring point in the corresponding simplified dynamic and static disk model gradually increases in the direction of decreasing radius as shown in Table 2. The characteristics of the flow field in the cavity will be inconsistent with the results of the conventional impeller disk model, especially the distributions of internal aerodynamic parameters and the number, position, and intensity of internal vortices. The flow characteristics of IBC are mainly circular shear flow and radial differential pressure flow. Generally, the airflow close to the impeller disk is centrifugal movement, and the airflow close to the casing disk is centripetal movement. The distributions of aerodynamic parameters are basically the same in the circumferential direction, and there are pressure and temperature gradient in the radial direction. The axial pressure gradient is negligible, but the axial temperature gradient exists because the impeller disk temperature is higher than the static wall. Figure 5

shows the measurement results of the dimensionless static pressure and static temperature distribution at different radii on the static wall surface of IBC under three typical conditions, including design conditions (34 kg/s), near-stall conditions (29 kg/s), and near-choke conditions (35 kg/s). The dimensionless static pressure or static temperature are the ratios of static pressure or static temperature experimental values and compressor inlet total pressure or total temperature experimental value, the specific formula expressions at each measuring point are as follows:

$$Cp_i = \frac{p_{s,i}}{p_{t,in}} (i = A, B, C, D, E, F, G, H) \tag{8}$$

$$CT_i = \frac{T_{s,i}}{T_{t,in}} (i = I, J, K, L, M, N, P, Q) \tag{9}$$

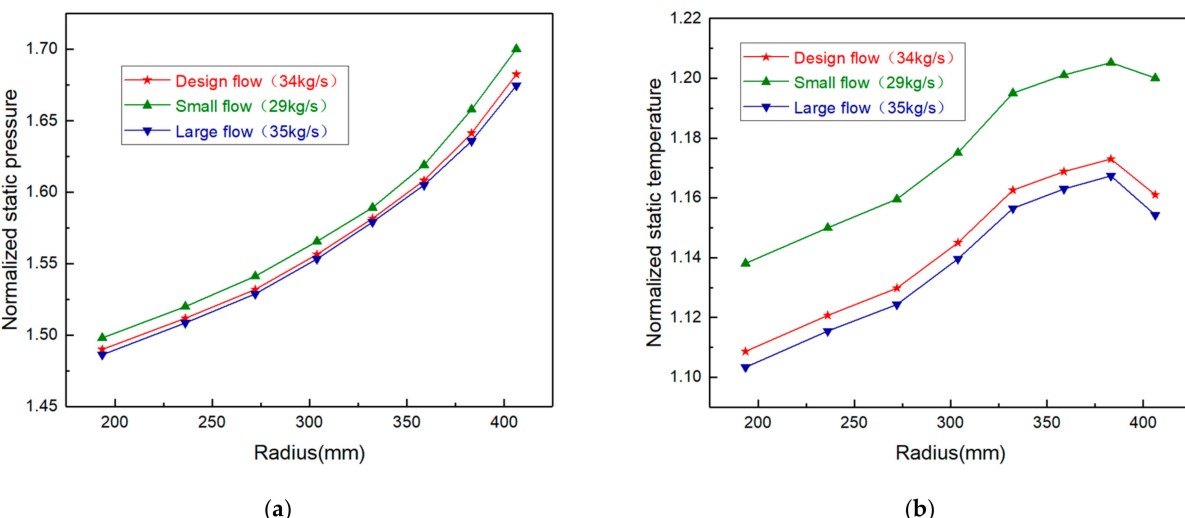

(**a**)  (**b**)

**Figure 5.** Distributions of aerodynamic parameters on the static wall of IBC: (**a**) static pressure distribution; (**b**) static temperature distribution.

As shown in Figure 5a, the static pressure on the static wall of the cavity under three typical conditions gradually decreases along the direction of decreasing radius. Because of the loss along the way of centripetal motion, the main sources are two aspects. Gas viscosity causes a large velocity gradient in the boundary layer along its thickness, and there is internal friction or viscous force between fluids, resulting in flow friction loss. Radial inward flow will generate tangential Coriolis force, causing the fluid to accelerate in the direction of rotation, and the relative speed of the direction of rotation will produce positive radial Coriolis force, which forms flow resistance loss together with centrifugal force.

It can be seen from Figure 5b that the static temperature on the static wall surface of the cavity under three typical conditions gradually rises along the direction of increasing radius but drops at the point near the coupling between the impeller outlet and the cavity inlet. The explanation for this phenomenon is as follows: the high pressure and lower temperature airflow at the impeller outlet brings the air-cooling effect, which gradually weakens along the decreasing radius; the viscous dissipation caused by the velocity gradient of the wall boundary layer along its thickness direction brings the effect of wind resistance and temperature rise, which gradually enhances along the increasing radius; the temperature rise effect is stronger in the small radius position, but the air-cooling effect is stronger in the large radius position, especially near the coupling position.

*4.2. Effect of Variable Operating Conditions*

The interface of the outer edge of the cavity is connected with the mainstream, which is the main channel for gas exchange between the compressor and the cavity. Due to

the influence of the airflow trail and direction at the impeller outlet, there are adjacent high-pressure and low-pressure areas on the pressure surface and the root of the trailing edge of the blade, which are different from other circumferential positions [8]. The axial velocity of the airflow in this region is much greater than the average locations, where the axial airflow exchange is strong. The mutual mixing and disturbance between low-entropy mainstream and high-entropy fluid in cavity mainly occur in there, causing the increase of the average entropy of mainstream, the loss of total pressure, the decrease of efficiency, and the increase of torque and shaft power. Because it is impossible to carry out an experimental study of the compressor without the IBC, the coupling characteristics are mainly explained by comparing the experimental value with the numerical calculation value.

Figure 6 shows the experimental characteristic curves of total pressure ratio, isentropic efficiency, torque, and shaft power under AVDs' design angle. The flow at the stall point and choke point is approximately 28.9 and 35.6 kg/s, and the flow at the highest efficiency point is approximately 31.5 kg/s. The highest efficiency is about 83.6%, and the highest total pressure ratio is about 2.35. The design point efficiency is about 81.5%, which is 2.5% away from the numerical efficiency of 83.6%; the design point total pressure ratio is about 2.25, which is 1.7% away from the numerical total pressure ratio of 2.29. Both the torque and the shaft power decrease with the decrease of the flow. The design point torque and shaft power are about 3723 N·m and 3374 kW, which are 2.5% and 2.4% away from the design point numerical values of 3631 N·m and 3292 kW, respectively. There are three main reasons for the above deviation: the design parameters do not consider the total pressure loss of IBC and AIGVs, etc.; there is a gap between the internal tongue and the wall of the inlet and outlet straight pipe pressure balance expansion joints, which also causes total pressure loss; the spiral airflow at the outlet of the compressor causes errors in the measurement of outlet parameters.

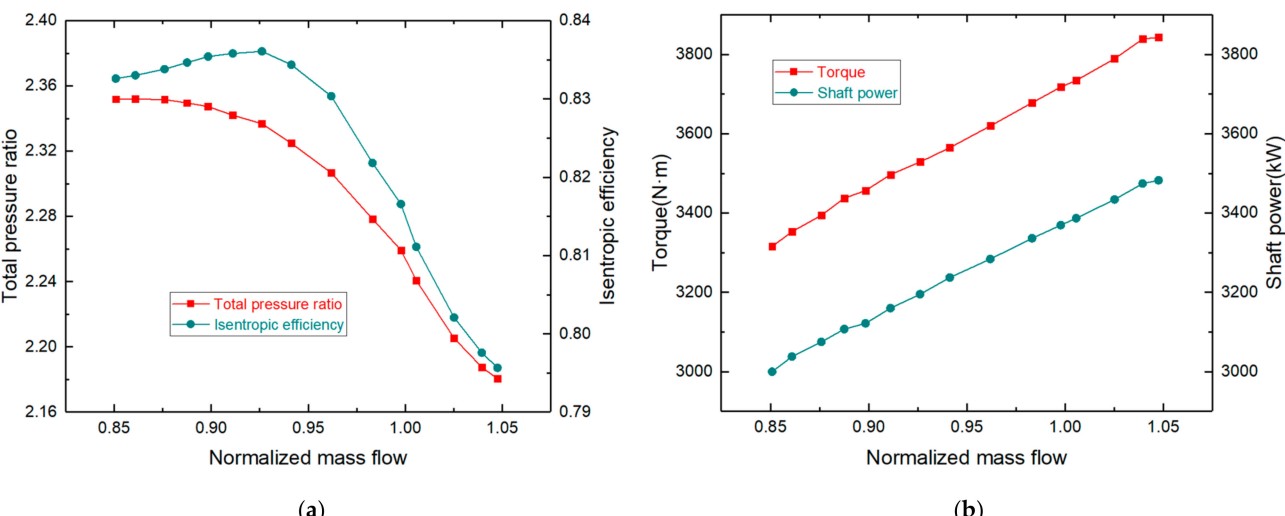

**Figure 6.** Characteristic curve of centrifugal compressor: (**a**) total pressure ratio and isentropic efficiency; (**b**) torque and shaft power.

Figure 7 shows the dimensionless static pressure or static temperature and respective loss characteristic curves at different radii on the static wall of IBC under AVDs' design angle. As described in 4.1, the static pressure of the cavity at each operating point in Figure 7a gradually decreases along the decreasing radius (H→A), and the static temperature of the cavity at each operating point in Figure 7b gradually rises along the direction of increasing radius (I→P) but drops at point (Q) near the coupling between the impeller outlet and the cavity inlet. The dimensionless static pressure loss in Figure 7c is the ratio of the static pressure loss from point H to point A, and compressor inlet total pressure experimental

value, which essentially reflect the pressure loss of airflow in the direction of centripetal movement along the decreasing radius, and the specific formula expressions are as follows:

$$Cp_{loss} = \frac{p_{s,loss}}{p_{t,in}} \left( p_{s,loss} = p_{s,H} - p_{s,A} \right) \tag{10}$$

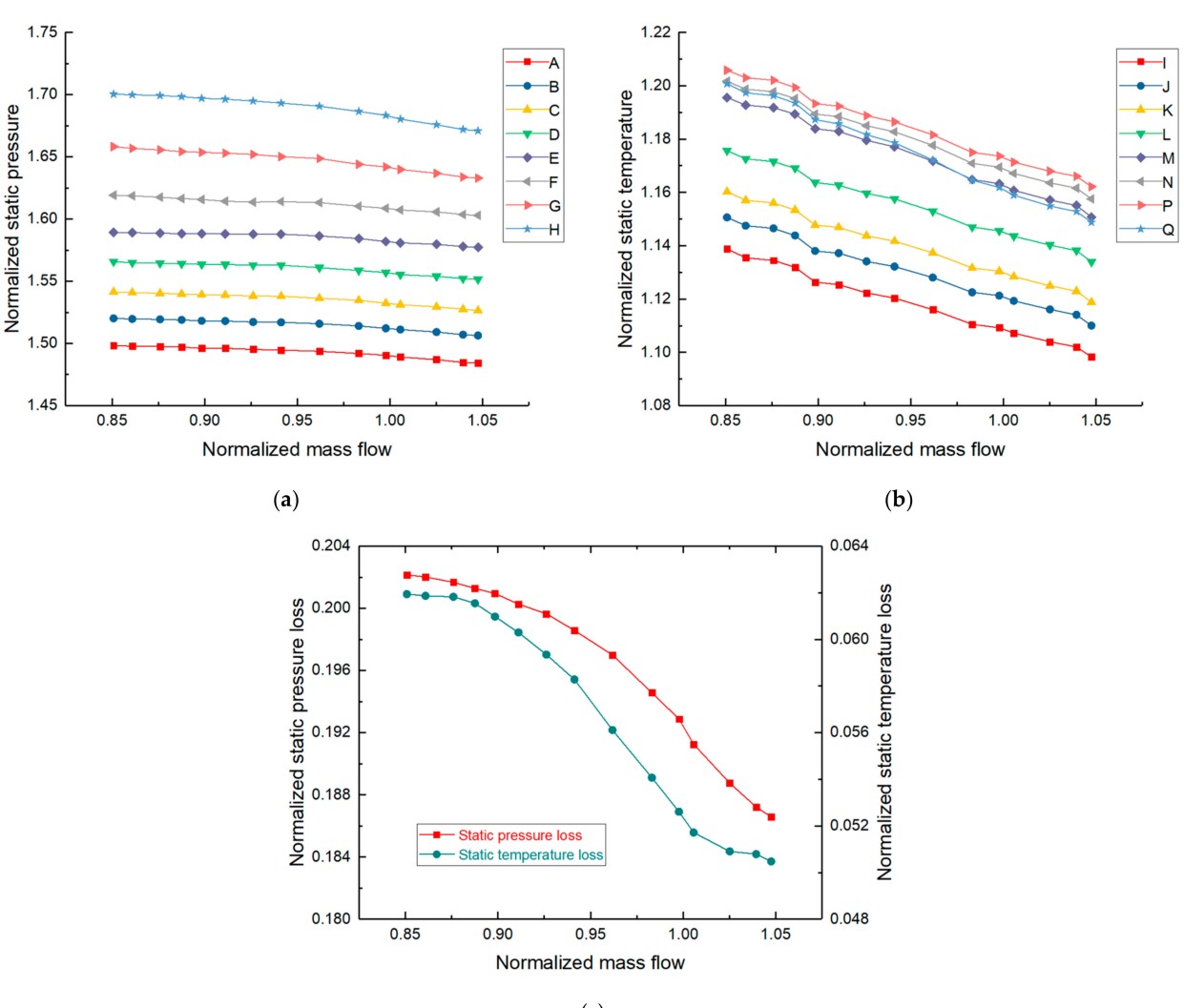

**Figure 7.** Characteristic curves of aerodynamic parameters and respective loss on the static wall of IBC: (**a**) static pressure; (**b**) static temperature; (**c**) static pressure loss and static temperature loss.

The dimensionless static temperature loss in Figure 7c is the ratio of the static temperature loss from point Q to point I, and compressor inlet total temperature experimental value, which essentially reflect the temperature loss of airflow in the direction of centripetal movement along the decreasing radius, and the specific formula expressions are as follows:

$$CT_{loss} = \frac{T_{s,loss}}{T_{t,in}} \left( T_{s,loss} = T_{s,Q} - T_{s,I} \right) \tag{11}$$

The value of the dimensionless static pressure loss in Figure 7c is much larger than the dimensionless static temperature loss, indicating that the pressure loss is more significant than the temperature loss along the way. The static pressure loss from point H to point A, and the static temperature loss from point Q to point I increase with the decrease of flow, that is, the radial static pressure and static temperature gradient increase as flow decreases.

In view of the above phenomenon, the following analyses are carried out in combination with the law of static pressure and static temperature change with flow at each position.

The static pressure at each position in Figure 7a increases with the decrease of the flow, and the large radius is more obvious than the small radius, resulting in the radial static pressure gradient increases as flow decreases. Refer to the analysis in Section 4.1, the decrease of flow will reduce friction loss and the influence of viscous force will be weakened, at the same time, the resistance effect of Coriolis force and centrifugal force will be increased, and the influence of inertial force will be enhanced. Both effects weaken the static pressure, but the effect of inertial force is stronger than that of viscous force at high speed [37]. The decrease in flow will enhance the effect of inertial force and increase the static pressure loss from point H to point A as shown in Figure 7c. The large radius is closer to the coupling than the small radius, which is more affected by the mainstream. The mainstream pressure for gas exchange increases with the decrease of the flow in Figure 6a, and the static pressure at the large radius will increase, but the static pressure loss will also increase along the way, resulting in the static pressure increase at the small radius is not as obvious as the large radius. Intuitively reflected in Figure 7a, the absolute value of the slope of the characteristic line at point A is smaller than that of the characteristic line at point H.

The static temperature at each position in Figure 7b rises with the decrease of the flow, and the large radius is more obvious than the small radius, resulting in the radial static temperature gradient increases as flow decreases. Refer to the analysis in Section 4.1, the decrease of flow will weaken the air-cooling effect because the flow entering the cavity decreases and the airflow temperature at the impeller outlet rises as shown in Figure 10 explained in detail in Section 4.3, simultaneously, the temperature rise of wind resistance effect is mainly affected by the speed gradient, which is affected little by the flow at high speed and large gap ratio [38]. The two effects weaken each other, but the decrease in flow mainly causes the weakening of the air-cooling effect, and the static temperature at each position rises. The large radius is closer to the incoming flow, the air-cooling effect is stronger, resulting in the air-cooling effect at point Q weaken more significantly and the static temperature rises more than other points. Intuitively reflected in Figure 7b, the absolute value of the slope of the characteristic line at point Q is larger than that of the characteristic line at point I.

*4.3. Effect of Adjustable Vaned Diffusers*

When the angle of AVDs changes, the coupling characteristics of the whole machine will be different, resulting in changes in the mainstream pressure and temperature of the gas exchange at the outer edge interface of the IBC. Although it is under the same rotating Reynolds number and mainstream flow, the flow coefficient and fluid pressure and temperature entering the cavity will be different. To study the effect of the angle change of AVDs on the internal flow field and aerodynamic parameters of the IBC, it needs to be based on the analysis of the coupling characteristics of compressor under different AVDs' angles.

Figure 8 shows the experimental characteristic curves of total pressure ratio, isentropic efficiency, torque, and shaft power under variable AVDs' angles ($-8°$, $-4°$, $0°$, $+4°$ and $+8°$). As the angle of AVDs decreases, the total pressure ratio and efficiency characteristic curves move to the lower left, and as the angle of AVDs increases, the total pressure ratio and efficiency characteristic curves mainly move to the right. Because the diffuser channel area under the negative diffuser angle is reduced, it is suitable for small flow conditions, and the compressor stall operating point moves to the left, which is opposite in the situation of the positive angle.

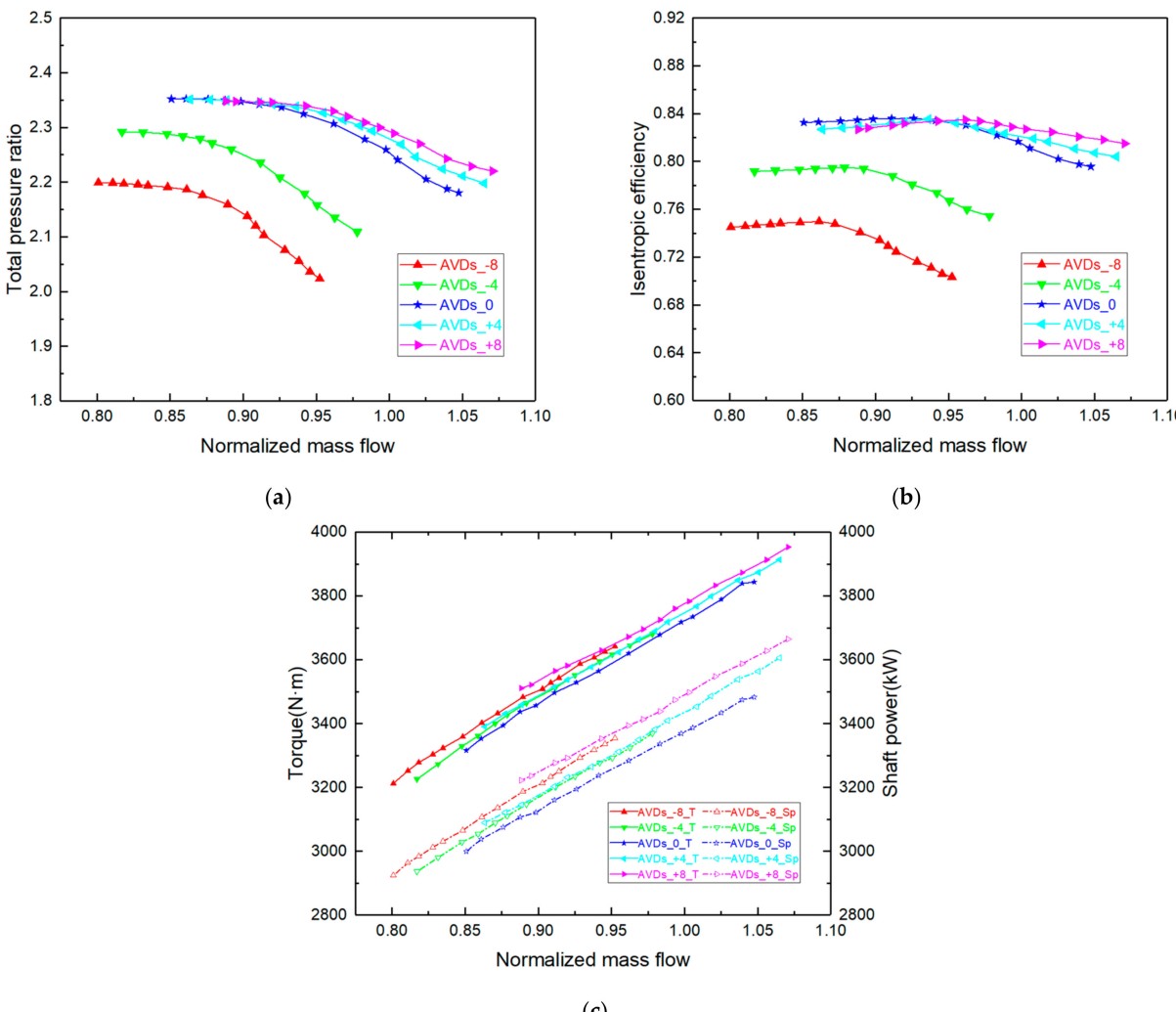

**Figure 8.** Characteristic curves of centrifugal compressor under variable AVDs' angles: (**a**) total pressure ratio; (**b**) isentropic efficiency; (**c**) torque and shaft power.

At a negative angle, as the degree of angle change increases, the blade inlet installation angle and the design inlet airflow angle are more mismatched, the greater the impact loss of the leading edge of the diffuser blade; and the smaller the tangential angle of the diffuser outlet airflow, making the airflow more difficult to discharge, the loss of the vaneless diffuser section increases; and the internal loss of the volute may also increase, which is much greater than that of the diffuser; the upstream and downstream effects of AVDs together cause the total pressure ratio and efficiency to decrease.

While at a positive angle, similar to the negative angle analysis, the impact loss of the leading edge of the diffuser blade increases with the degree of angle change; but the larger the tangential angle of the diffuser outlet airflow, making the airflow easier to discharge, the loss of the vaneless diffuser section is decreased, and the internal loss of the volute may also be decreased; the combined effects of the upstream and downstream of this AVDs lead to no significant drop in the total pressure ratio and efficiency.

When the angle of AVDs decreases or increases, the torque and shaft power at the same flow both increase slightly in Figure 8c, and the change intensifies with the increase of the angle change. Because the higher the degree of mismatch between the blade inlet installation angle and the design inlet airflow angle, the greater the impact loss of the leading edge of the diffuser blade, and at the same time, it induces that the matching degree between the upstream impeller and AVDs becomes worse, resulting in more power consumption.

Figure 9 shows the dimensionless static pressure and its loss characteristic curves at different radii on the static wall of IBC under variable AVDs' angles (−8°, −4°, 0°, +4° and +8°). Under the negative angle of AVDs such as in Figure 9a,b, the static pressure characteristic curve at each position moves to the upper left. Because the air impact point (maximum static pressure point) at the leading edge of the blade under the negative angle moves to the suction surface of the diffuser blade, the static pressure of the gap between the impeller outlet and the diffuser inlet increases locally, and the static pressure in the cavity increases integrally [52], which is opposite in the situation of the positive angle such as in Figure 9e,f. To sum up, the static pressure at each position decreases with the increase of AVDs' angle. As the angle of AVDs increases, the static pressure loss characteristic curve from point H to point A of the IBC moves to the upper right in Figure 9d. Because with the increase of AVDs' angle, the distance between the front edge of AVDs blade and the outer edge of the IBC is shortened, the coupling relationship between AVDs and IBC is closer, and the airflow is diverted from the impeller outlet to the diffuser channel in advance, resulting in the amount of gas exchange at the outer edge of the cavity is reduced. Although the mainstream flow remains unchanged, but the flow into the cavity decreases. Combining the analysis in Section 4.2, it can be inferred that the static pressure loss from point H to point A of the IBC increases with the increase of the inlet installation angle of AVDs, and increases with the decrease of the mainstream flow under each AVDs' angle.

The law of static pressure changes with the mainstream flow at each position under variable AVDs' angles is more complicated. Under the zero angle in Figure 9c, it increases with the decrease of the flow and the large radius is more obvious than the small radius as described in Section 4.2. When the angle of AVDs is positive such as in Figure 9e,f, the trend of static pressure at each position is the same as the zero angle, but the magnitude of the rise increases more significantly. When the angle of AVDs is negative such as in Figure 9a,b, the static pressure at each position even decreases with the decrease of the flow and the small radius is more obvious than the large radius. The explanation for this phenomenon is as follows. Under the negative angle, although the static pressure in the cavity increases integrally as mentioned in the previous paragraph, the static pressure of the gap between the impeller outlet and the diffuser inlet decreases significantly as the flow decreases, the same as the static pressure at point H of the IBC, which is opposite in the situation of the positive angle. Combined with the law that the static pressure loss from point H to point A increases as the flow decreases in Figure 9d, it is not difficult to find that the small radius changes significantly than the large radius under the negative angle.

Figure 10 shows the experimental characteristic curves of the dimensionless static temperature at point Q of the IBC and impeller outlet temperature under variable AVDs' angles (−8°, −4°, 0°, +4° and +8°). Both characteristic lines move to the upper left as the diffuser angle decreases, and move to the upper right as the diffuser angle increases. The temperature difference between them drops with the increase in the degree of change in the angle of AVDs, leading to the air-cooling effect is weakened.

Figure 11 shows the dimensionless static temperature and its loss characteristic curves at different radii on the static wall of IBC under variable AVDs' angles (−8°, −4°, 0°, +4° and +8°). Under the negative angle of AVDs such as in Figure 11a,b, the static temperature characteristic curve at each position moves to the upper left, and under the positive angle of AVDs such as in Figure 11e,f, the static temperature characteristic curve at each position moves to the upper right. The static temperature at each position rises with the increase of the change degree of AVDs' angle. Because the air-cooling effect is weakened but the effect of wind resistance and temperature rise is basically unaffected. As the angle of AVDs increases, the static temperature loss characteristic curve from point Q to point I of IBC moves to the upper right in Figure 11d. Similar to the analysis of the static pressure loss from point H to point A of the IBC, although the mainstream flow is unchanged, but the flow in the cavity decreases. Combining the analysis in Section 4.2, it can be inferred that the static temperature loss from point Q to point I of IBC increases as AVDs' angle increases and increases as the mainstream flow decreases under each AVDs' angle.

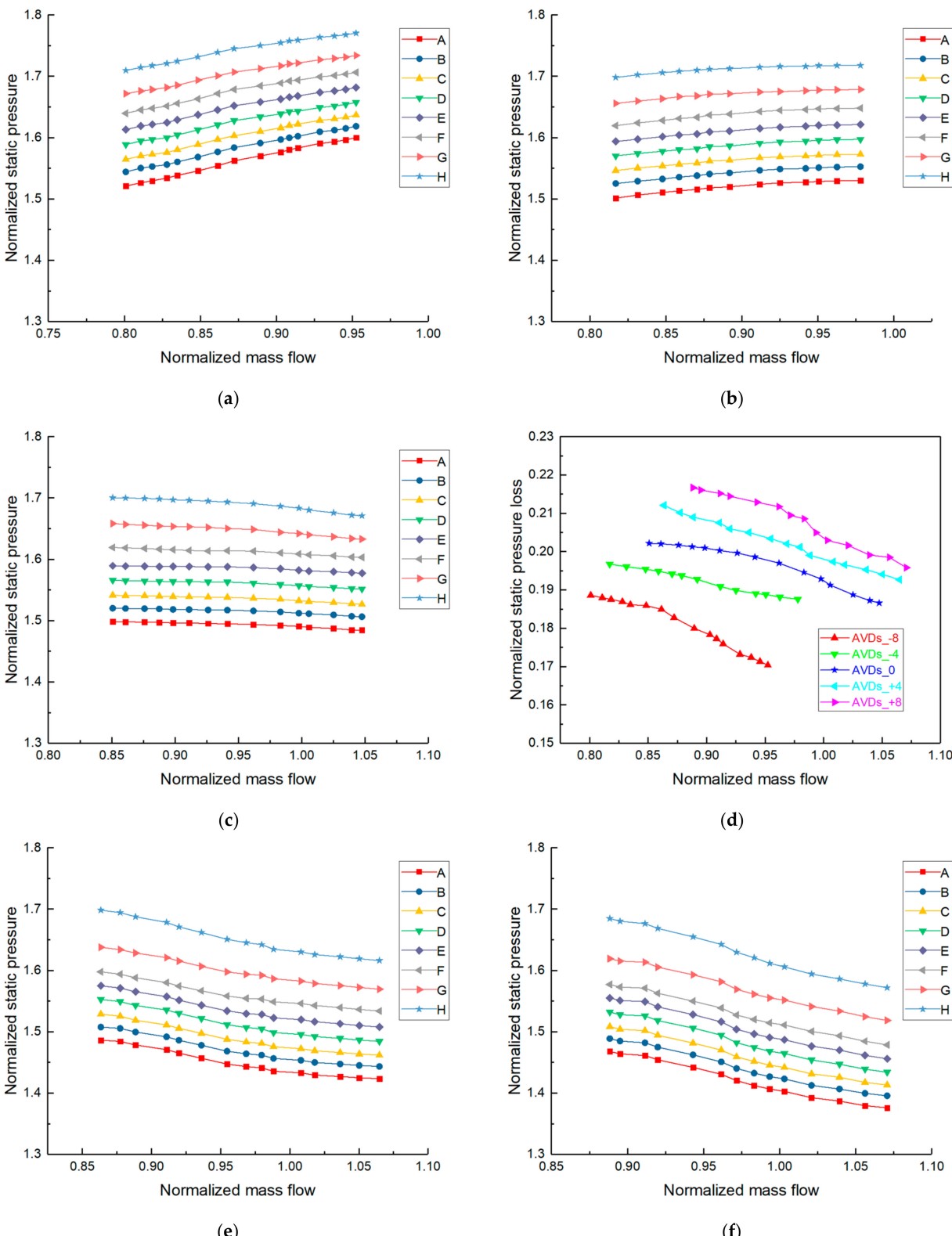

**Figure 9.** Characteristic curves of static pressure and its loss on the static wall of IBC under variable AVDs' angles: (**a**) static pressure with −8° AVDs; (**b**) static pressure with −4° AVDs; (**c**) static pressure with 0° AVDs; (**d**) static pressure loss with variable AVDs' angles; (**e**) static pressure with +4° AVDs; (**f**) static pressure with +8° AVDs.

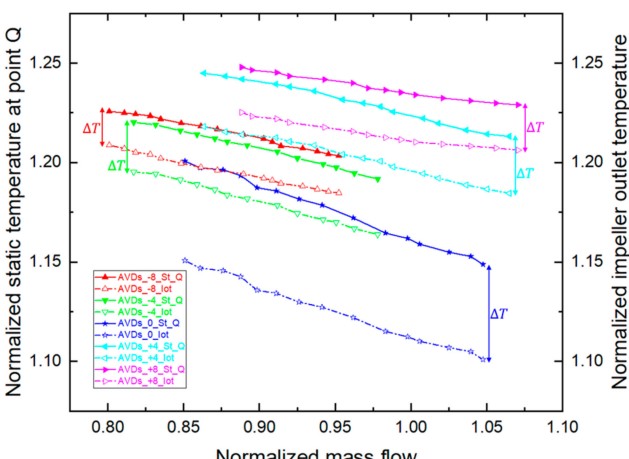

**Figure 10.** Characteristic curves of static temperature at point Q of IBC and impeller outlet temperature under variable AVDs' angles.

(**a**)

(**b**)

(**c**)

(**d**)

**Figure 11.** *Cont*.

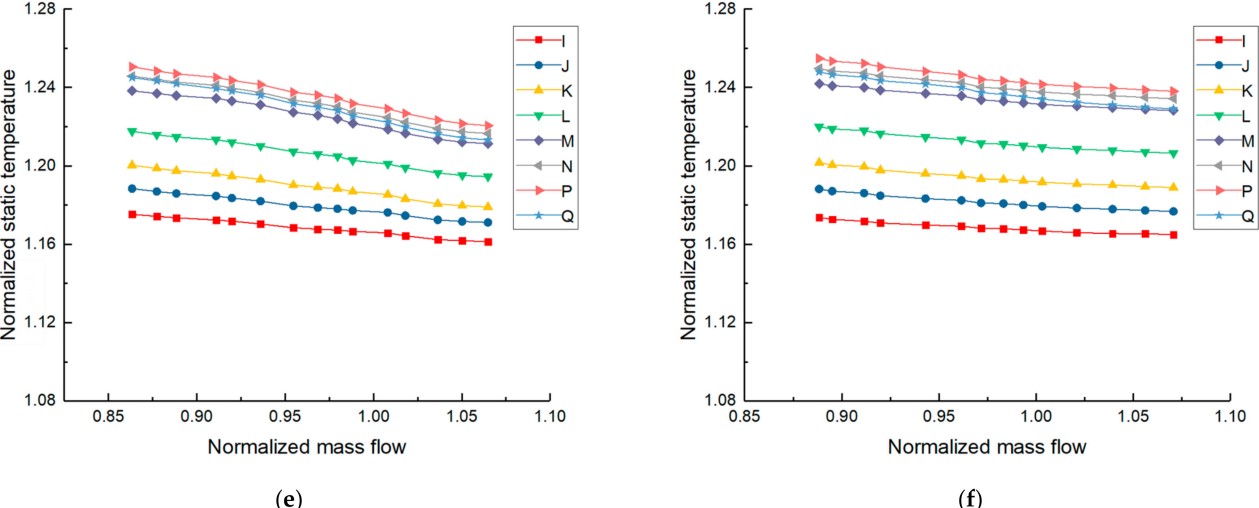

**Figure 11.** Characteristic curves of static temperature and its loss on the static wall of IBC under variable AVDs' angles: (**a**) static temperature with −8° AVDs; (**b**) static temperature with −4° AVDs; (**c**) static temperature with 0° AVDs; (**d**) static temperature loss with variable AVDs' angles; (**e**) static temperature with +4° AVDs; (**f**) static temperature with +8° AVDs.

The law of static temperature changes with the mainstream flow at each position under variable AVDs' angles is more complicated. Under the zero angle in Figure 11c, it rises as the flow decreases and the large radius is more obvious than the small radius as described in Section 4.2. When the angle of AVDs is positive or negative such as in Figure 11a,b,e,f, the trend of static temperature at each position is the same as the zero angle, but the magnitude of the rise increases not significantly with the increase of the angle change. The explanation for this phenomenon is as follows. Under the positive or negative angle, although the static temperature in the cavity both rise as mentioned in the previous paragraph, the static temperature of impeller outlet temperature both rise not significantly as the flow decreases in Figure 10, the same as the static temperature at point Q of the IBC. Combined with the law that the static temperature loss from point Q to point I increases as the flow decreases in Figure 11d, it is not difficult to find that the obvious degree of increase in the large radius compared to the small radius is reduced.

## 5. Conclusions

The aerodynamic parameters distributions in the cavity of the semi-open impeller disk were measured in detail, relying on the large-scale CAES multi-stage intercooling centrifugal compressor closed test rig. The flow field law in cavity and coupling characteristics with IBC under different mainstream flow and variable AVDs' angles were analyzed, which can guide its weight reduction design optimization strategy to reduce leakage and windage losses. Through the processing and analysis of the test data, the results can be summarized as follows.

(1) The static pressure on the static wall of the IBC gradually decreases along the direction of decreasing radius, and the loss along the way of centripetal motion mainly includes friction loss and the resistance loss of Coriolis force and centrifugal force. The static temperature on the static wall of the IBC gradually rises along the direction of increasing radius but drops near the coupling between the impeller outlet and the cavity inlet, because the air-cooling effect is stronger than the effect of wind resistance and temperature rise in there.

(2) Under AVDs' design angle, the static pressure at each position on the static wall of the IBC increases with the decrease of the flow, and the static temperature at each position rises as the flow decreases. The large radius is more obvious than the small radius, resulting in the radial static pressure and static temperature gradient both increase with the decrease of flow.

(3) The static pressure at each position on the static wall of IBC decreases with the increase of the inlet installation angle of AVDs, the static temperature at each position rises with the increase in the degree of change in AVDs' angle. The static pressure loss and static temperature loss in the direction of decreasing radius both increases as the AVDs' angle increases and increases as the mainstream flow decreases under each AVDs' angle.

(4) Under the zero angle of AVDs, the static pressure and static temperature at each position both increase as the flow decreases and the large radius is more obvious. Under the positive angle, the trend of static pressure is the same as the zero angle, but the magnitude of the rise increases more significantly; under the negative angle, it even decreases with the decrease of the flow and the small radius is more obvious. When the angle of AVDs is positive or negative, the trend of static temperature is the same as the zero angle, but the magnitude of the rise increases not significantly as the change degree of angle increases.

(5) Under the same impeller speed, changing the mainstream flow or ADV's angle actually just only changes the flow, pressure, and temperature of the gas exchange at the outer edge interface, which indirectly affects the flow field in the IBC.

**Author Contributions:** Conceptualization, Z.L.; methodology, Z.L.; software, Z.L.; validation, Z.L., W.G. and J.S.; formal analysis, Z.L.; data curation, Z.L.; writing—original draft preparation, Z.L.; writing—review and editing, Z.Z. and Q.L.; supervision, H.C.; funding acquisition, H.C. All authors have read and agreed to the published version of the manuscript.

**Funding:** This research was funded by National Science Fund for Distinguished Young Scholars (51925604), the International Partnership Program, Bureau of International Co-operation of Chinese Academy of Sciences (182211KYSB20170029), the Guizhou Province Large Scale Physical Energy Storage Technology Research and Development Platform ([2019]4011), the Guizhou Province Large Scale Physical Energy Storage Engineering Research Center program ([2017]951).

**Institutional Review Board Statement:** Not applicable.

**Informed Consent Statement:** Not applicable.

**Conflicts of Interest:** The authors declare no conflict of interest.

## Nomenclature

| | |
|---|---|
| $m$ | Mass flow (kg/s) |
| $n$ | Rotating speed (r/min) |
| $p_s$ | Static pressure (Pa) |
| $p_t$ | Total pressure (Pa) |
| $P_s$ | Shaft power (W) |
| $PR_t$ | Total pressure ratio |
| $T$ | Toque (N·m) |
| $T_s$ | Static temperature (K) |
| $T_t$ | Total temperature (K) |
| $\eta_{is}$ | Isentropic efficiency |
| **Greek letter** | |
| $\gamma$ | Isentropic exponent |
| $\eta$ | Efficiency |
| **Dimensionless number** | |
| $Cp$ | Dimensionless static pressure |
| $Cp_{loss}$ | Dimensionless static pressure loss |
| $CT$ | Dimensionless static temperature |
| $CT_{loss}$ | Dimensionless static temperature loss |
| $C_w$ | Dimensionless mass flow |

| | |
|---|---|
| $G$ | Gap ratio |
| $Re_\omega$ | Rotational Reynolds number |
| $\beta_0$ | Inlet swirl ratio |
| $\lambda_T$ | Turbulent flow parameter |

**Subscript**

| | |
|---|---|
| $i$ | Marked measuring point |
| $in$ | Compressor inlet |
| $is$ | Isentropic |
| $out$ | Compressor outlet |
| $s$ | Static state or Shaft |
| $t$ | Stagnation state |

**Abbreviation**

| | |
|---|---|
| AIGVs | Adjustable Inlet Guide Vanes |
| AVDs | Adjustable Vaned Diffusers |
| CAES | Compressed Air Energy Storage |
| IBC | Impeller Backside Cavity |
| RPM | Revolutions Per Minute |
| VFD | Variable-Frequency Drive |

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
