# Peer review of "Experimental Study on Effects of Adjustable Vaned Diffusers on Impeller Backside Cavity of Centrifugal Compressor in CAES"

_energies, doi:10.3390/en14196187_

Round 1

Reviewer 1 Report

This study by the Zhihua Lin et al., reports on the effects of adjustable vaned diffusers on impeller backside cavity of centrifugal compressor in compressed air energy storage (CAES). The authors have carried out thorough and systematic investigation by measuring the static pressure and static temperature at distinctive radii on the static wall of the impeller backside cavity (IBC) and established a coupling relationship between IBC and the centrifugal compressor. This work was further supported by identifying the variations in the distributions of aerodynamic parameters in the cavity under various mainstream flows. These findings are likely to be of interest to the community in developing compressed air energy storage systems with improved energy conversion efficiency for distributed energy applications. Overall, this an interesting work with motivation and contributions stated clearly by identifying and addressing a problem that has not been studied yet. Therefore, the reviewer did not find any issues with the paper and feels that it can be accepted by the journal in the present form.

Author Response

Thank you for your patient and careful review. You have made a detailed summary of the motivation and contribution of this article, thank you again for your high recognition of this article and direct acceptance of the comments.

Reviewer 2 Report

In this paper the effect of adjustable vaned diffuser on impeller backside cavity of a centrifugal compressor has been analyzed through an experimental test facility. The topic is very interesting, in particular to better understand the flow behavior due to the back cavity; the paper is quite well organized with a good quality in the complex. However some suggestions to further improve the quality are reported below.

The introduction is quite complete, but a greater literature background on numerical works could be provided. In particular to have a more attractive paper it is possible to add some references also for off-limits condition, by providing a particular flow mechanism at the diffuser inlet (near the back cavity inlet). In particular I suggest to add for example these refs.:

  • Cravero, C.; Marsano, D. “Criteria for the stability limit prediction of high-speed centrifugal compressors with vaneless diffuser. Part I: flow structure analysis”, Proceedings of the ASME Turbo Expo 2020: Turbomachinery Technical Conference and Exposition, Virtual Conference, 21-25 September 2020, ASME paper GT2020-14579.
  • Cravero, C.; Marsano, D. “Criteria for the stability limit prediction of high-speed centrifugal compressors with vaneless diffuser. Part II: the development of prediction criteria”, Proceedings of ASME Turbo Expo 2020: Turbomachinery Technical Conference and Exposition, Virtual Conference, 21-25 September 2020, ASME paper GT2020-14589.

These are two parts paper that propose some criterion to detect the stability limit in centrifugal compressor, by highlighting the region of the diffuser inlet how one of the most important surge causes especially for low speed.

The geometrical informations of the compressors are provided, but a zoom figure is required for the back cavity region; is it possible to provided geometrical information also for it (for example the gap between the rotor disk and the fixed wall)?

The experimental method is well described

The results section is quite good organized and well described with some diagrams to parametrically analyze the flow mechanism. However some legends are not very clear (can you enlarge and choice better colors?). Is it possible to provide also some flow visualizations?

The conclusions are well summarized, but you could add also some numerical values.

Author Response

Thank you for your patient and careful review. Your comments and suggestions are valuable and helpful for revising and improving our paper, as well as the important guiding significance to our researches.

Point 1: A greater literature background on numerical works could be provided.

Response 1: Thank you for your advice. We have added those two papers as references, which propose some criterion to detect the stability limit in centrifugal compressor, by highlighting the region of the diffuser inlet how one of the most important surge causes especially for low speed.

Point 2: A zoom figure is required for the back cavity region; is it possible to provided geometrical information also for it (for example the gap between the rotor disk and the fixed wall).

Response 2: Thank you for your comment. We have added a detailed zoom map of the locations of the measuring points, which provide the radius and axial distance of the gap at each point.

Point 3: The experimental method is well described.

Response 3: Thank you for your high evaluation. We have sorted out the experimental method description in more detail.

Point 4: Some legends are not very clear (can you enlarge and choice better colors?). Is it possible to provide also some flow visualizations?

Response 4: Thank you for your patient review. In response to the legend display problem you mentioned, we have changed the color and thickness of some of the lines, hoping to meet your requirements. Regarding the process visualization you mentioned, there are too many measurement points, it is difficult for us to perform a perfect process visualization analysis, only the visualization of the results can be displayed, I hope you understand it with magnanimity.

Point 5: The conclusions are well summarized, but you could add also some numerical values.

Response 5: Thank you for your suggestion. Due to space and other issues, we have elaborated the numerical part in another article, and compared it with the test results, which will form a complete article of a higher level. I hope to get your comments in the future again.

Special thanks to you for your good comments and suggestions.

Reviewer 3 Report

General

The paper addresses the experimental study on the effect of Adjustable Vane Diffusers (AVD) applied to largescale centrifugal compressors on the overall efficiency and the main aerodynamic parameters (pressure and temperature) in the Impeller Backside Cavity (IBC). The work is well motivated. According to the comprehensive literature survey presented, there is a lack of experimental data in this field. Therefore, the aim was to collect data in order to create guidelines for the optimal design.  The authors completed successfully this work: appropriate test rig was equipped and numerous experimental data of the static pressure and temperature distribution in IBC zone were obtained.  The accuracy of the results was estimated. Comments and explanation of the phenomena observed are given when necessary, as well. The conclusions are systematized in groups and truly reflect the results.

The paper is interesting and provides a valuable information.  It has enough merits to be published in the Journal after minor revision according to the remarks listed below:

Remarks and recommendations; Orthography and typing errors

  • Line 167: Equation numeration must be (2), not (1).
  • Line 168: Eq. (3) – What is “PS”?; please add to Nomenclature.
  • Line 176: Eq. (6) – What is “his”?, perhaps you mean “ηis”, please amend.
  • Lines 237 – 246: Two very long consecutive sentences, please edit.
  • Nomenclature: Dimensions are missing.
  • Figures: Somewhere dimensions are missing.
  • Add dimensions where necessary.

Author Response

Thank you for your patient and careful review. You have made a detailed summary of the motivation and contribution of this article. Your comments and suggestions are valuable and helpful for revising and improving our paper, as well as the important guiding significance to our researches.

Point 1: Line 167: Equation numeration must be (2), not (1).

Response 1: Thank you for your advice. We have corrected the equation number in the revised draft.

Point 2: Line 168: Eq. (3) – What is “PS”?; please add to Nomenclature.

Response 2: Thank you for your comment. We have added it to the nomenclature with an extended explanation.

Point 3: Line 176: Eq. (6) – What is “his”?, perhaps you mean “ηis”, please amend.

Response 3: Thank you for your suggestion. We have modified it in the revised draft.

Point 4: Lines 237 – 246: Two very long consecutive sentences, please edit.

Response 4: Thank you for your patient review. Two very long continuous sentences have been split into two sentences to introduce dimensionless pressure loss and temperature drop respectively.

Point 5: Nomenclature: Dimensions are missing. Figures: Somewhere dimensions are missing. Add dimensions where necessary.

Response 5: Thank you for your careful review. We have added dimensions description where the full text needs to be added.

Special thanks to you for your good comments and suggestions.

Round 2

Reviewer 2 Report

The paper is now ready for the publication. All my questions and suggestions have been answered or added in the revised paper.